# Molecular Biomarkers of Glioma

**DOI:** 10.3390/biomedicines13061298

**Published:** 2025-05-26

**Authors:** Punsasi Rajakaruna, Stella Rios, Hana Elnahas, Ashley Villanueva, David Uribe, Sophia Leslie, Walaa A. Abbas, Larissa Barroso, Stephanie Oyervides, Michael Persans, Wendy Innis-Whitehouse, Megan Keniry

**Affiliations:** 1School of Integrative Biological and Chemical Sciences, The University of Texas Rio Grande Valley, 1201 W. University Dr., Edinburg, TX 78539, USA; punsasi.rajakaruna@utrgv.edu (P.R.); stella.rios01@utrgv.edu (S.R.); hana.elnahas01@utrgv.edu (H.E.); ashley.villanueva03@utrgv.edu (A.V.); david.uribe01@utrgv.edu (D.U.); sophia.leslie01@utrgv.edu (S.L.); walaa.abbas01@utrgv.edu (W.A.A.); larissa.barroso01@utrgv.edu (L.B.); stephanie.oyervides01@utrgv.edu (S.O.); michael.persans@utrgv.edu (M.P.); 2Burrell College of Osteopathic Medicine, 3501 Arrowhead Dr, Las Cruces, NM 88001, USA; winnis@burrell.edu

**Keywords:** glioma, glioblastoma, IDH, oligodendroglioma, astrocytoma

## Abstract

In this review, we discuss how mutations in glioma are associated with prognosis and treatment efficacy. A fascinating characteristic of glioma and all cancers is that while common growth and developmental pathways are altered, the characteristic mutations are distinct depending on the specific type of tumor with concomitant prognoses. Next-generation sequencing, precision medicine, and artificial intelligence are boosting the employment of molecular biomarkers in cancer diagnosis and treatment. Understanding the biological underpinnings of distinct mutations on critical signaling pathways is crucial for developing novel therapies for glioma.

## 1. Introduction

Glioma encompasses many types of brain tumors originating from glial cells, such as astrocytes, oligodendrocytes, and ependymomas [1,2,3]. Glioblastoma accounts for 50% of primary, malignant brain tumors and has a median survival of 12–14 months [1]. Glioblastoma is highly infiltrative, invading neighboring structures, and vascularized [1]. Histologically related malignant brain tumors that harbor mutations in *IDH1*/*IDH2* were formerly described as glioblastoma, but since the 2021 World Health Organization classification of central nervous system tumors, they are now denoted as *IDH*-mutant astrocytoma (even if high-grade) or oligodendroglioma (*IDH*-mutant with 1p/19q co-deletion) [1,2,3]. This review discusses progress in developing glioma molecular biomarkers that inform prognosis and treatment strategies.

## 2. *IDH1* and *IDH2* Mutations Are Crucial Biomarkers for Glioma

A seismic shift in glioblastoma classification occurred based on the molecular biomarkers *IDH1* (*isocitrate dehydrogenase 1*) and *IDH2* (*isocitrate dehydrogenase 2*). Mutations in *IDH1*/*IDH2* (abbreviated *IDH*) were found in 10–12% of brain tumors that were histologically classified as glioblastoma [4,5]. The *IDH* mutations were associated with a much better prognosis than non-mutant *IDH* glioblastoma [4]. Mutations associated with *IDH1* and *IDH2* ultimately led to the reclassification of gliomas, changing them from being glioblastomas to astrocytomas (even if high grade) and oligodendrogliomas by the World Health Organization (WHO) in 2021 [1,3]. In the presence of *IDH* mutations, a hypermethylation phenotype is observed [6]. IDH1 normally catalyzes the conversion of isocitrate to alpha-ketoglutarate. The IDH1-R132H mutant protein catalyzes alpha-ketoglutarate conversion to D-2-hydroxyglutarate (D-2-HG) in an NADPH-dependent manner [7]. The accumulation of D-2-HG alters metabolism (such as the Krebs cycle and glycolysis) and blocks the demethylation of histones and nucleotides, resulting in the silencing of tumor suppressor genes and halting cellular differentiation [6,7]. Similar impacts on the Krebs cycle and gliomagenesis were found with *FH* and *SDH* mutations, as shown in Figure 1 [8,9].

### 2.1. The Frequency of IDH Mutations in Glioma

Mutations in the *IDH1* and *IDH2* genes are present in glioma at varying rates. In the cases analyzed, *IDH1* mutations were detected in 10–15%, and *IDH2* mutations were 1–5% [4,5,10]. Lower-grade gliomas, such as grades II and III, had a higher *IDH1/2* mutation frequency of 70–80% [4,5,10]. Alternatively, wildtype *IDH* (primary glioblastomas) showed a low occurrence of these mutations [11,12]. This significant difference highlights the important role that *IDH* mutations play in identifying distinct types of gliomas.

### 2.2. IDH Mutations Associated with Immature Glial Differentiation Programs

*IDH* mutations are found in younger patients and are associated with tumors with a specific genetic profile [13]. *IDH* mutations are typically of the proneural glioma subtype, associated with the glioma-CpG island methylator phenotype (G-CIMP) [6,14]. At a gene expression level, *IDH*-mutant gliomas harbor programs that resemble the recycling of early glial developmental programs, favoring either the early stages of astrocytic or oligodendrocyte cell fate specification [15,16]. Interestingly, single-cell sequencing revealed that very similar populations of cells were found in *IDH*-mutant astrocytoma (*IDH*-A) and oligodendroglioma (*IDH*-O). Both have a population of undifferentiated, proliferative stem-like cells, as well as non-proliferative populations resembling astrocyte and the oligodendrocyte lineages [15,16].

### 2.3. IDH Mutation and Prognosis in Glioma

Observations indicate that patients with *IDH* mutations have more extended survival periods than glioma patients with the wildtype *IDH* gene [4,14]. Yamauchi et al. (2018) describe the median progression-free survival (PFS) for *IDH* mutants as 4.7 years, with a 5-year life expectancy rate of 42%. In glioblastoma (without these mutations), patients had a PFS of 1.4 years and a 5-year life expectancy rate of 14% [17]. Metellus et al. (2010) indicated that *IDH* mutations were associated with prolonged patient survival and therapeutic response [18]. This finding shows that *IDH* mutation status is an essential predictor of glioma prognosis. Circulating cell-free DNA (cfDNA) can be used in liquid biopsies to detect *IDH1* mutations in glioma patients (especially *IDH1*-R132H) [19,20,21,22]. This mutational information has a significant impact on the tumor classification of oligodendroglioma, and of *IDH* mutant astrocytoma or glioblastoma with associated prognoses. The National Comprehensive Cancer Network (NCCN) considers *IDH* mutations important in assessing risks and treating gliomas. The NCCN recommends testing for these mutations as a standard diagnostic procedure [22]. Further studies on prognostic indicators in *IDH*-mutant glioma are underway. Recently, it was found that copy number variations and methylation changes were associated with deregulated miRNA expression in *IDH*-mutant glioma [23]. The knockdown of *miR-155-5p*, *miR-196a-5p*, *miR196b-5p*, *miR-200a-3p*, *miR-503-5p*, and *miR-15b-5p* significantly reduced the expression of immune evasion proteins PD-L1, CTLA4, and FOXP3 [23].

### 2.4. Association Between Drug Efficacy and IDH Mutation

There are multiple research avenues exploiting *IDH* mutations and drug efficacy. *IDH* mutants influence how tumors react to specific treatments [24,25,26]. Patients’ responses to treatments with temozolomide (TMZ) change with the presence or absence of *IDH* mutations [24,25,26]. Although TMZ is an effective treatment for gliomas with *IDH* mutations, determining the best dosage to use and understanding the tumor response to TMZ is complex. New strategies that use TMZ in combination with other drugs may be needed for more effective outcomes [25]. This emphasizes that *IDH* mutation is a potential factor in tailoring patient treatment plans. *IDH* mutations are initial factors in glioma development, leading to the buildup of D-2-hydroxyglutarate (D-2-HG), which alters the energy usage of cells and affects DNA within cells. These are crucial factors in understanding how tumors behave and react to therapy [26].

The NOA16 clinical trial examined the *IDH1* peptide vaccine in patients with *IDH1*-R132H mutation; see Table 1. A 64% three-year progression-free survival (PFS) rate and an overall survival rate (OS) of 84% were reported, indicating the success of immunotherapy methods [27]. The *IDH* inhibitors, ivosidenib, which targets *IDH1*, and vorasidenib, which targets *IDH1* and *IDH2*, showed a favorable response in individuals with *IDH1*-mutant Grade IV astrocytoma; see Table 1 [27]. This suggests that targeted *IDH* inhibitors are valuable in treating specific tumors [24]. Research is ongoing about how *IDH* mutations affect survival. Further information is needed from ongoing clinical studies to draw more conclusions. As we gain knowledge about the molecular mechanisms behind these mutations, personalized treatment strategies and effective care for patients become more feasible.

## 3. Molecular Markers on the PI3K Pathway in Glioma

The PI3K/AKT pathway is crucial in cell growth, survival, and metabolism. When bound by a ligand, receptor tyrosine kinases (RTKs) become activated, leading to a cascade of signaling events. Phosphatidylinositol 3-kinase (PI3K) becomes activated by RTKs, which then phosphorylates phosphatidylinositol (4,5)-bisphosphate (PIP2) to become the second messenger, phosphatidylinositol (3, 4, 5)-trisphosphate (PIP3) [4,5,46]. Afterward, PIP3 recruits AKT (protein kinase B) to the cellular membrane, which becomes phosphorylated and activated by phosphoinositide-dependent kinase, PDK1 [46,47].

The PI3K/AKT pathway promotes cellular growth and proliferation in part via the downstream mTOR complexes, mTORC1, and mTORC2. The activation of mTOR begins by the phosphorylation of tuberous sclerosis complex 2 (TSC2) by AKT, leading to cytosol sequestration [46,47]. This sequestration hinders the ability of TSC2 to interact with Rheb. Next, Rheb accumulates at the lysosome, which then activates mTORC1, allowing for the regulation of anabolic processes such as the synthesis of proteins, lipids, and nucleotides [46,47].

### 3.1. PTEN (Phosphatase and TENsin) Loss of Function Mutations in Glioblastoma

*PTEN*, or Phosphatase and TENsin homolog, is a tumor suppressor gene located on chromosome 10q23, encoding a dual specificity phosphatase (403 amino acids) that down-regulates the PI3K pathway by diminishing PIP3 [48]. PTEN induces apoptosis, hinders cancer proliferation, and promotes genomic stability [28]. In glioblastoma, *PTEN* mutations are frequently found in primary tumors, at approximately 32%, compared to secondary tumors (4%) [49,50]. Wang et al. analyzed 34 primary glioblastoma cases, observing *PTEN* mutations in 44% (15 out of 34) and 74% (25 out of 34), exhibiting LOH at the *PTEN* locus [25]. Additionally, 60% of primary glioblastomas that exhibited LOH carried a somatic *PTEN* mutation [51].

*PTEN* mutation is associated with the mesenchymal subtype of glioblastoma and a poor prognosis due to an increase in tumor aggressiveness, proliferation, and treatment resistance [52,53]. There is a significantly lower survival rate in patients who present with *PTEN* mutations (62.5% 36-week survival rate) compared to those who do not present (92.9% 36-week survival rate) [54]. The *PTEN* mutation incidence significantly differs between WHO Grade III and Grade IV glioma classifications (*p* = 0.024) [46]. *PTEN* null mutations induce the PI3K/AKT pathway, which can drastically reduce the efficacy of targeted treatments and chemotherapy [29]. Tumors with *PTEN* mutations respond less favorably to standard therapies such as TMZ [29].

Deficiency in *PTEN* disrupts cell cycle checkpoints, allowing damaged cells to mutate further and avoid apoptosis [29]. This disruption of cell cycle mechanisms contributes to treatment resistance [30]. The *BCL2* family regulates intrinsic apoptotic pathways and includes anti-apoptotic members, *BCL2*, *BCL-xL*, *MCL1*, and pro-apoptotic members, *BAX* and *BAK* [30]. *PTEN*-deficient cells with elevated levels of anti-apoptotic proteins, such as *BCL2*, resist TMZ-induced apoptosis [30,31].

Currently, *PTEN* mutations can be identified preoperatively via MRI. Through gene ontology analysis, researchers have discovered that radiomic signatures were significantly associated with processes commonly seen in *PTEN* mutations [51]. The transcriptome data of 64 patients obtained from the Cancer Genome Atlas were analyzed in a Pearson correlation test to select genes that were significantly related to radiomic signature—of which 200 genes were found to be significantly positively correlated [55]. The gene ontology analysis revealed that the radiomic signature was significantly associated with *PTEN*-related biological processes such as cell migration, cell motion, and anti-apoptotic [39]. These results imply that radiomic signatures can predict *PTEN* mutation status non-invasively, as the underlying genetic characteristics of the tumor can be reflected [56].

In another study, data from 244 glioma patients were collected from a local database (n = 77) and the Cancer Imaging Archive (n = 167), and they were randomly divided into a training set (n = 170) and a validation set (n = 74) [57]. Three models were developed to determine the prediction of *PTEN* mutations in patients: a convolutional neural network (CNN) model, a radiomics model, and an integrated model (radiomics and CNN) [57]. Of the three models, the integrated model showed the highest accuracy (86.5%) compared to the CNN (81.1%) and radiomics (66.2%) model [57].

*PTEN* mutations were detected via liquid biopsy through circulating cell-free DNA (cfDNA) and tumor DNA (tDNA) in GBM patients [58]. A total of 75 samples (25 GBM blood samples, 25 GBM tumor samples, and 25 healthy control blood samples) were studied; the cfDNA characteristics were compared between GBM patients and the control group [58]. *PTEN* mutations were found in 33% of tumor samples and 30% of cfDNA samples, suggesting that liquid biopsy could serve as a non-invasive diagnostic tool for GBM [58].

### 3.2. PIK3CA-Activating Mutations in Glioma

Phosphatidylinositol 3 Kinase (PI3K) is a heterodimer composed of a catalytic subunit, such as the *p110a* isoform, encoded by *PIK3CA*, which converts phosphatidylinositol 4,5 bisphosphate (PIP2) into phosphatidylinositol 3,4,5 trisphosphate (PIP3), and a regulatory subunit, p85, encoded by the *PIK3R1* (*phosphatidylinositol-4,5-bisphosphate 3-kinase regulatory subunit alpha*) gene. The regulatory subunit stabilizes the catalytic subunit and is critical for maintaining various cellular processes, including cell growth, survival, and proliferation. *PIK3CA* mutations have been reported in between 5 and 30%; this variation in the range can be attributed to different methodologies for analyzing mutation frequency in glioblastoma [32]. *PIK3CA* is mutated in hotspots on exons 10 and 21, encoding the constitutively active mutants H1047R, E542K, and E545K [32]. These activating mutations trigger an aberrant downstream signaling cascade of the AKT/mTOR pathway, which increases cell proliferation, survival, and contributes to oncogenesis.

Studies have shown that *PIK3CA* mutations are observed in tumors with concurrent mutations in the isocitrate dehydrogenase (*IDH1* and *IDH2*) genes [32]. *IDH*-mutant glioma had a mutation rate of 9% compared to *IDH* wildtype glioblastoma, which had a mutation rate of 3% [32]. *IDH1*-mutant glioma is associated with the proneural subtype and a better prognosis. *PIK3CA* mutations expressed in *IDH* wildtype tumors had shorter overall survival [32].

Mutations in *PIK3CA* contribute to resistance mechanisms via the β-catenin and Hedgehog signaling pathways [32]. The selective PI3K inhibitor Alpelisib is being investigated to deter constitutive *PIK3CA* mutations in cancer [33]. Additionally, combination therapies that target the PI3K/AKT pathway in conjunction with MEK or EGFR inhibitors can provide synergistic effects compared to monotherapies [33,34]. Preclinical studies combining PI3K and MEK inhibitors showed effectiveness in cancer models with *RAS* mutations in basal-like breast cancer. One study found synergistic effects in *K-RAS*-mutant cancer models with the PI3K inhibitor, GDC-0941, and the MEK inhibitor, GDC-0973. This synergistic effect is also reflected in glioblastoma models, where an orthotopic mouse model treated with both GDC-0941 and GDC-0973 displayed substantial tumor reduction [35,36]. Additionally, the combination of these inhibitors effectively targets critical components in the PI3K/AKT and EGFR/MEK/ERK pathways, resulting in enhanced apoptosis, reduced tumor growth, and bypassing compensatory activation. *PIK3CA*-mutant tumors treated with PI3K/AKT and EGFR/MEK/ERK dual inhibition displayed significant regression in experimental mouse models; see Table 1 [35,37].

Recently, researchers suggested that *PIK3CA* mutations can be detected preoperatively via dual-time-point [18F]FET PET/CT imaging [59]. Patients aged 18 and older suspected to have a primary brain tumor were recruited between June 2021 and January 2024 (n = 76) [59]. Of these patients, 51 had *IDH* wildtype glioblastoma, 13 had *IDH*-mutant astrocytoma, and 12 had *IDH*-mutant oligodendroglioma [59]. Of the 13 astrocytoma patients, 2 were classified as Grade II, 6 were classified as Grade III, and 5 were classified as Grade IV. In contrast, the oligodendroglioma patients were evenly distributed between Grades II and III [59]. Two PET/CT scans were performed after [18F]FET injection, the first 20 min post-injection and the second 80 min post-injection [59]. The maximum standardized uptake value (SUV) was measured for each scan, with results showing that patients with *PIK3CA* mutations had a significantly higher max SUV compared to patients with no mutations; the difference remained significant even after an adjustment for MRI enhancement, tumor grade, pathology type, and age [59].

Additionally, *PIK3CA* mutations can be detected through liquid biopsy [58]. Palande et al. found that *PIK3CA* mutations were detected in 9% of tumor samples and 7.8% of cfDNA samples [58]. These results offer a promising non-invasive method for preoperatively detecting *PIK3CA* mutation status in glioma patients.

### 3.3. PIK3R1 Mutations in Glioblastoma

*PIK3R1* encodes alternatively spliced isoforms of the regulatory subunit involved in the Class IA PI3K lipid kinase heterodimer; p85a is the most commonly expressed regulatory subunit variant from the *PIK3R1* gene [60]. The p85a regulatory subunit associates with the catalytic subunit p110, encoded by *PIK3CA*, to stabilize and, to some degree, inhibit kinase activity, leading to a low level of PI3K pathway activity. In cancer, the p85a subunit is frequently reduced in expression or mutated, leading the catalytic subunit p110 to associate with another regulatory subunit p85b, encoded by the gene *PIK3R2*, leading to high levels of PI3K lipid kinase activity [60,61,62,63].

*PIK3R1* mutation rates vary within the context of glioblastoma, but according to the Cancer Genome Atlas, the mutation frequency of *PIK3R1* in glioma is 9.9% [62]. Mutations in *PIK3R1* alter the regulatory subunit’s ability to interact with p110α, resulting in aberrant hyperactivation of the AKT/mTOR signaling pathway, leading to uncontrolled cell growth and survival [60,61,62]. Within the iSH2 domain, mutations such as DKRMNS560del, R574fs, and T576del have been discovered to have a gain-of-function mutation, showing increased kinase activity of the PI3K complex, which results in increased independent phosphorylation of PIP2 to PIP3, activating AKT signaling [61]. Furthermore, truncating mutations, such as R348* and L370fs, can localize into the nucleus, activating other pathways, such as JNK, thus promoting tumor progression and bypassing apoptosis [38]. Additionally, loss-of-function mutations within *PIK3R1* have been shown to impair the inhibitory regulation of the catalytic subunit p110α and even disrupt the ability to stabilize *PTEN* [40,61]. Mutations such as E160* impair *PTEN* stability, resulting in hyperactivation of the AKT signaling pathway [40]. Research has found a connection between specific glioblastoma subtypes and *PIK3R1* mutations, with most observed in the mesenchymal subtype [64]. *IDH* wildtype gliomas have also shown a high frequency of *PIK3R1* mutations [65]. Mutations in *PIK3R1* are almost mutually exclusive with *PIK3CA* mutations and often coexist with other mutations, such as *TERT* promotor mutations and/or *PTEN* mutations [65].

*PIK3R1* mutations are associated with poor prognosis due to their disruption of crucial signaling pathways, such as the PI3K/AKT pathway and Wnt/β-catenin [64]. These signaling pathways become hyperactivated, thus increasing tumor invasiveness and aggressiveness. Patients presenting with *PIK3R1* mutations often have poor progression-free survival (PFS) and OS [60]. One study that identified potential markers correlated to poor prognosis in diffuse glioma patients validated that patients presenting with a *PIK3R1* mutation had an overall poorer prognosis than individuals with *PIK3R1* wildtype tumors, with a median survival of 2.4 years and 5.4 years, respectively [60,66]. Another study found that *PIK3R1* mutation was associated with poor prognosis in the mesenchymal subtype of glioblastoma. These characteristics result from the *PIK3R1* mutation, leading to constitutive activation of PI3K signaling [64].

Studies have shown that tumors harboring *PIK3R1* mutations have an increased sensitivity to PI3K pathway inhibitors, such as the AKT inhibitor MK2206, as shown in Table 1 [61,62]. This increased sensitivity poses the potential to build on and create targeted therapeutic approaches. Furthermore, therapeutic combinations using PI3K inhibitors in conjunction with EGFR inhibitors have shown synergistic efficacy [61]. Overall, *PIK3R1* mutations trigger a hyperactive PI3K/AKT pathway, leading to increased cellular proliferation, resistance to apoptosis, and enhanced migration and invasion, allowing metastasis and tumor cells to invade surrounding tissues [67].

*PIK3R1* mutations can be detected via liquid biopsy, as 7.8% of mutations were found in tumor samples and 7.0% were found in cfDNA samples [58]. Although these mutations were detected less frequently in comparison to *PTEN* and *PIK3CA* mutations, the detection rate of *PIK3R1* mutations was similar in both tumor and cfDNA samples (only a 0.8% difference), which implies that these mutations can be reliably detected when present in GBM patients [58].

## 4. Biomarkers on the RAS Pathway in Glioma

The RAS pathway is aberrantly activated by mutations in cancers, including glioblastoma [68]. The molecular circuitry of the RAS pathway includes the binding of ligands to receptor tyrosine kinases such as Epidermal Growth Factor Receptor (EGFR) or Platelet-Derived Growth Factor Receptor (PDGFR), which leads to GTP binding/activation of the G-protein RAS. RAS-GTP activates serine/threonine kinase BRAF, which phosphorylates MEK. Then, MEK phosphorylates MAPK kinases to promote growth, survival, and loss of contact inhibition to promote tumorigenesis [69]. RAS signaling also impacts the cytoskeleton and other targets, such as the PI3K pathway [70].

The mutation rates of RAS pathway components vary, depending on the gene. The three RAS encoding genes *N-RAS*, *H-RAS,* and *K-RAS* are mutated at a low frequency in glioblastoma (approximately 2% of cases) [71]. The occurrence of *BRAF* mutations is about 3% in adult glioblastoma [72,73]. *EGFR*, *NF1*, and *PDGFRA* mutations are more frequent in glioblastoma [74]. *EGFR* mutations are over-expressed in glioblastoma tumors (50–60% of the time) [74,75]. *PDGFRA* mutations are common in glioblastoma; one study found that 40% of glioblastomas had *PDGFRA* mutations [76]. *NF1* is mutated in approximately 13% of glioblastoma [55]. Interestingly, *ATRX* mutations are associated with Ras pathway mutations in glioblastoma; *ATRX* mutation is associated with a better prognosis [77].

RAS pathway mutants are differentially associated with glioblastoma subtypes. *EGFR* amplification/mutation is strongly associated with the classical subtype of glioma [14,41]. *NF1* mutations are primarily associated with the mesenchymal glioblastoma subtype [55,78], and *PDGFRA* is correlated with the proneural glioma subtype [14,79]. Additionally, *BRAF* mutations are common in epithelioid glioma [80,81,82].

There is evidence that *EGFR* amplification or mutation is associated with poor prognosis. *EFGR* amplification was detected by fluorescent in situ hybridization (FISH) [68]. To analyze the *EFGR* copy number status, SNP Array analysis was used by extracting high-molecular-weight genomic DNA from glioblastoma patients [68]. Patients with *EGFR* alteration had the shortest survival rate, supporting a contribution to poor prognosis [83]. *PDGFRA* was detected using next-generation sequencing (NGS) by extracting DNA from formalin-fixed paraffin-embedded (FFPE) tumor samples [69]. However, these results were obtained after they performed gross or subtotal tumor removal on 56 patients and partial tumor removal (PTR) or biopsy on 51 patients, including treating them with temozolomide during radiotherapy [69]. Their defined levels of biomarker gain were 3–5 copies, and amplification was greater than five copies [69]. The median overall survival of patients with and without *PDGFRA* gain-of-function mutations/amplification was 15.2 and 29.5 months, which showed a significantly improved prognosis for patients without the *PDGFR* gain-of-function mutations/amplification [84].

*EGFR* mutation and expression are crucial in drug efficacy for treating glioblastoma [42,85,86,87]. Studies have examined several small molecular kinase inhibitors and antibodies to target EGFR, including Afatinib, Dacomitinib, Osimertinib, Rociletinib, Gefitinib, Erlotinib, and Lapatinib, as well as the monoclonal antibody Cetuximab [42,86,87]. However, the main drawbacks of these drugs are that they cannot outcompete ATP, and the heterogeneity in *EGFR* mutations successfully affects their efficacy, feedback mechanisms, low specificity, and increases side effects (diarrhea) [78,86,87]. Researchers demonstrated that the expression of *PDGFRA* is associated with drug efficacy in treating glioblastoma [88]. The experiment was performed on *Nod.Scid.Il2rg*^0^ (NSG) mice using HIF1α inhibitor delivered by intracranial implantation [88].

*BRAF* and *NF1* mutations correlate with poor prognosis [89,90]. DCC 0.1.0 Software was used to detect circRNAs (cirBRAFs) obtained from the Illumina HiSeq Sequencer [87]. A protocol called Volcano Plot filtering was used to identify differentially expressed circRNAs [76]. *NF1* mutations were significantly associated with poor prognosis in glioblastoma [55,72]. In this investigation, researchers assembled a retrospective cohort of 542 *IDH*-wildtype GBM to study detected *NF1* mutations [37]. *NF1* immunohistochemistry was used to identify *NF1* alterations [37]. Two NF1 antibodies were used for this technique, which were NFC, Sigma-Aldrich, St. Louis, MO, USA (recognizes the last 281 amino acids of human neurofibromin), and iNF-07E, iNFixion (recognizes amino acids 863–867 of *NF1*) [37]. Bioscience combination therapies that target multiple steps of the RAS signaling pathway are under study to develop new interventions [91].

### The Capicua Transcriptional Repressor as a Glioma Biomarker on the Ras Pathway

The Capicua transcriptional repressor (*CIC*) gene encodes for a large protein [92] that represses downstream RTK/RAS/ERK signaling [93]. The *CIC* gene is located at the cytoband 19q13.2 and encodes a high-mobility group-box transcriptional repressor. Under normal circumstances, the CIC protein represses proliferation genes [92,94]. *CIC* mutations lead to protein inactivation, which results in upregulation of the CIC target genes, activation of proliferative pathways, inhibition of differentiation, and a statistically significant poorer outcome in patients with co-deletion 1p/19q [92]. Chromosome 19q13.2 has somatic mutations, comprising insertions/deletions encompassing the *CIC* gene, which has been validated by deep sequencing in recurrent oligodendrogliomas [56].

A study researching *CIC* inactivating mutations sequenced DNA from 127 oligodendroglial tumors and found *CIC* mutations in 47% of the patients, with 59 out of 60 also having *IDH* mutations. It was also noted during this sequencing that chromosomes 4, 9p, and 10q were often lost [92]. Another study characterizing *CIC* mutations hypothesized that because *CIC* mutations were highly associated with *IDH* mutation, the hemizygous presentation of mutant *CIC* may be correlated with oligodendroglioma pathogenesis in *IDH* mutants [56]. NGS, IHC, and WTS studies performed on *CIC*-mutated formalin-fixed paraffin-embedded (FFPE) tissue samples showed that mutations in *CIC* resulted in an increased rate of 1p/19q co-deletion, *TERT* promoter mutations, *MGMT* promoter methylation, and deficient mismatch repair resulting in microsatellite instability [94]. The Catalogue of Somatic Mutations in Cancer (COSMIC) showed a total mutation rate of 7.69% of the *CIC* gene in all central nervous system cancers [95]. Most *CIC* mutations occur in the 1p/19q co-deletion, associated with a worse prognosis, but when disregarding the co-deletion cases, *CIC* mutations in astrocytoma and oligodendroglioma were 2% [56,92]. A study that classified glioma samples according to the 2021 WHO CNS5 criteria showed that *CIC* had a mutation rate of 52.1% in oligodendroglioma, 1.7% in astrocytoma, and 0.6% in glioblastoma [94].

A common form of mutation to the *CIC* gene is an unbalanced translocation resulting in the loss of one copy of chromosome 1p and 19q. This occurs in up to 80% of oligodendrogliomas and 15% of oligoastrocytomas. Most co-deleted gliomas are also mutated on *IDH1*/*IDH2*, which is associated with a better prognosis than *IDH* wildtype [92].

Work on Hs683 cells found that a *CIC* mutation resulted in a defect in the nuclear targeting of this protein [92]. These mutations occur homogeneously in the coding sequence and have been seen upstream and in the protein/protein interaction domain [92]. Most truncation mutations resulted in the loss of the nuclear localization sequence (NLS) and the C1 domain, with the rest only involving the loss or partial loss of the C1 domain.

Despite many advances in cancer treatment, glioma remains challenging to treat, partly due to its inherent characteristics, such as location and aggressiveness, but also because of the natural defense of the blood–brain barrier. Many researchers have begun to dissect molecular mechanisms and gene therapies that may bring a better prognosis. Some of these new research areas are non-invasive, such as radiomic techniques that blend magnetic resonance images with a program scanning 11 features that can predict *CIC* mutation in glioma with an accuracy of 94.2% at a 95% confidence interval [96]. Circulating tumor DNA (ctDNA) from patients’ blood can also be evaluated for the *CIC* co-deletion biomarker with 55% sensitivity and 100% specificity via microsatellite analysis [97]. The standard of care for glioblastoma usually includes resection and treatment with TMZ, an alkylating agent [94]. Treatment with TMZ in patients with *CIC* mutation showed an increased median overall survival rate of 1874 days versus 602 days [94]. Besides the current treatment protocols, studies have shown that MEK/MAPK inhibitors are viable treatment options for malignancies with MAPK dysfunctions, such as gliomas with *CIC* mutations [94]. Trametinib, binimetinib, selumetinib, and cobimetinib are four inhibitors already approved for use in adult and pediatric solid tumors [94]. The observed degradation of CIC in RAS/ERK-driven tumors may increase ERK inhibitors’ efficacy for treating glioma [93]. Another method of gene therapy targeting the deletion of the ERK binding interface for CIC stability in vivo may be a subject of further investigation for the extended survival of patients [93].

## 5. *TP53* as a Glioma Biomarker

The *Tumor Suppressor 53* (*TP53*) gene on chromosome 17 encodes the p53 protein. This gene is critical in controlling cell division, DNA repair, apoptosis (programmed cell death), and cell cycle arrest [98,99]. It is an important part of the regulatory network involved in tumorigenesis. It can be functionally categorized into several areas: stemness, cell metabolism, inflammatory responses, the tumor microenvironment, and the immune response [100].

Mutations in the *TP53* gene have been strongly linked to cancer cell proliferation and metastasis, with over 50% of human tumors expressing mutant *TP53* [98]. *TP53* mutations occur in approximately 30% of primary glioblastoma cases, 65–90% of secondary glioblastoma cases, and 46.7% of pediatric glioma cases [95,101]. Many of the mutations mentioned above occur in the DNA-binding domain, a prominent hotspot for *TP53* mutations [102].

*TP53* mutation causes altered sensitivity to 27 known drugs, including MIRA-1, a drug that targets TP53 through the p53 pathway; Dabrafenib, which targets the protein kinase BRAF through the ERK MAPK signaling pathway [103], and Selumetinib, which targets MEK1 and MEK2 mitogen-activated protein kinases through the ERK MAPK signaling pathway [103].

The knockdown of mutated *TP53* in glioblastoma cells leads to a five-fold increase in chemosensitivity to the drug TMZ due to O^6^-methyltransferase (MGMT) expression. Reduced expression of mutant *TP53* leads to decreased MGMT expression [104]. The IC_50_ of temozolomide in T98G cells was 575.32 μg/mL before mutant *TP53* silencing and 112.15 μg/mL after. In U138 cells, the IC_50_ was 510.79 μg/mL before knockdown and 127.79 μg/mL after [104]. Since MGMT counteracts the effects of TMZ by repairing TMZ-induced DNA damage, it can be concluded that the chemosensitivity of glioblastoma to TMZ decreases with mutant *TP53* and increases with the knockdown of mutant *TP53* [104]. In another study, Kaplan–Meier survival analysis of 512 glioma patients (248 with TP53 alterations, 264 without) showed that the *TP53* altered group had a shorter median survival of 79.99 months as compared to the *TP53* unaltered group with a median survival of 95.57 months, indicating that *TP53* alterations are associated with poorer overall survival [100]. *TP53* status is significantly associated with disease progression and patient survival in glioblastoma during radio- and chemotherapy [105,106].

*TP53* mutation status is currently evaluated through blood tests, bone marrow tests, or tumor tissue biopsies and it can be incorporated into diagnostic panels for glioma classification. Although not yet widely implemented in liquid biopsy for glioma, *TP53* mutations are detectable in circulating tumor DNA in other cancers, suggesting future potential [107]. The WHO CNS5 recognizes *TP53* alterations as common in both primary and secondary glioblastoma, and *TP53* mutation status may impact therapeutic decision making, for example, regarding temozolomide sensitivity and clinical trial eligibility [3].

## 6. *LRP1B* Is a Key Tumor Suppressor in Glioma

The *LDL receptor-related protein 1B* (*LRP1B*) gene encodes a member of the low-density lipoprotein (LDL) receptor family [108]. LRP1B has a wide range of biological functions, from cargo transport to cell signaling. It is among the top 10 altered genes in human cancer, suggesting it plays a vital role as a tumor suppressor gene [108].

Inactivation of LRP1B promotes cell migration and invasion, primarily through the activation of the RhoA/Cdc42 signaling pathway or the upregulation of urokinase plasminogen activator receptor (uPAR), a receptor involved in extracellular matrix degradation [109]. This activation leads to the reorganization of the actin cytoskeleton, which is essential for cell movement. In glioblastoma, approximately 4% of cases harbor mutations in *LRP1B*.

Research indicates a strong link between *LRP1B* deletion and the development of glioma, particularly in patients with wildtype *IDH1/2* [110]. Studies reveal that 10% of *IDH1/2* wildtype glioblastoma tissues contain *LRP1B* deletions, significantly reducing expression in tumor samples compared to healthy tissue. *LRP1B* deletion has been strongly associated with poor prognosis in glioblastoma patients [109].

Furthermore, mutations in *LRP1B* are linked to altered sensitivity to specific chemotherapeutic drugs, including AZD4877, which targets EG5, and MK-1775, which inhibits WEE1 and PLK1 [102].

Although *LRP1B* mutations are not yet routinely evaluated in standard clinical practice, they can be identified through next-generation sequencing (NGS) panels performed on biopsy specimens. LRP1B’s association with poor prognosis in *IDH*-wildtype glioblastoma shows its emerging potential as a prognostic biomarker and a possible therapeutic target for drugs.

## 7. The *SMARCB1* Gene as a Glioma Biomarker

The *SMARCB1* gene, also known as SWI/SNF (*SWItch/Sucrose Non-Fermentable-related Matrix-associated Actin-dependent Regulator of Chromatin subfamily B member 1*), encodes a protein subunit of the SWI/SNF protein complexes. These complexes regulate gene expression through chromatin remodeling [111]. This process alters nucleosome positioning on the DNA, affecting gene activity during expression.

SWI/SNF complexes participate in various cellular processes, including DNA repair, replication, and cell growth regulation. As a part of these complexes, the SMARCB1 protein is believed to function as a tumor suppressor, helping to prevent uncontrolled cell growth. Research indicates that SMARCB1 suppresses *CCND1* transcription and inhibits CDK 4/6 activity by binding to and recruiting HDAC in the G1 phase of the cell cycle, ultimately affecting cell cycle progression [112,113].

Although mutations in the *SMARCB1* gene are found in only 2–3% of glioblastoma cases, studies suggest these mutations are linked to an earlier disease onset [102,113]. This indicates that SMARCB1 may play a critical role in glioma’s early development and presentation. Additionally, *SMARCB1* mutations are associated with altered sensitivity to one known drug, Axitinib, which targets receptors PDGFR, KIT, and VEGFR [95].

While the relationship between *SMARCB1* mutations and glioblastoma prognosis is not well researched, *SMARCB1* loss has been identified in several rare pediatric and adult cancers, termed *SMARCB1*-deficient cancers, most of which have a poor prognosis [114].

*SMARCB1* mutations can also be detected using NGS assays on tumor tissue [115]. While SMARCB1 is not yet a formal requirement for glioma classification in WHO CNS5, a loss of SMARCB1 function is increasingly recognized in pediatric and early-onset gliomas [113]. It is being investigated as a marker for prognosis and therapeutic vulnerabilities, particularly for resistance mechanisms related to cell cycle regulation [116].

## 8. *TERT* Mutations Drive Telomerase in Glioma

*Telomerase reverse transcriptase* (*TERT*) is one of the top genes mutated in glioblastoma. Mutations in the promoter region of the *TERT* gene occur in 70–80% of all glioblastomas [117,118,119]. Higher-grade glioma tissues were seen to have more *TERT* mutations than low-grade glioma [117]. *TERT* promoter mutations result in a gain of function of TERT protein production, increasing telomere length and resulting in immortality due to the cell dividing but not undergoing senescence [120]. One study found that in 35 of the 37 glioblastoma cell lines studied, there was either a C228T or C250T promoter mutation [119]. These *TERT*-mutated cell lines had elevated levels of telomerase expression, but the amount of telomerase produced varied between cell lines [119]. Furthermore, in another study, in 55% of the 358 glioblastomas analyzed, *TERT* promoter mutations (C228T, C250T) were detected; of the 55%, 73% had a C228T mutation, and 27% had a C250T mutation; only one glioblastoma had both mutations [2]. Within that study, it was found that mutations in the *TERT* promoter occurred significantly more often in *IDH1* wildtype glioblastomas (187 out of 322; 58%) compared to *IDH1*-mutated gliomas (10 out of 36; 28%; *p* = 0.0056) [39,118].

Glioblastoma patients with *TERT* mutations had a shorter survival period and an overall poorer prognosis than those without the mutations. A study concluded that the *TERT* mut is an independent factor for poor prognosis [121]. In a study, 359 plasma samples were taken from 110 glioma patients, and by using droplet digital polymerase chain reaction (ddPCR), the DNA was isolated and analyzed for three key gene mutations, including *TERTp*. It was found that the *TERTp* C228T mutation was detected in 88% of the plasma ctDNA. Therefore, detecting *TERTp* by ddPCR demonstrates the potential use of the technique for glioma liquid biopsy, indicating a method of preoperational detection [122].

TERT may be a chemotherapeutic target for glioblastoma. Specifically, mycophenolic acid (MPA) is a known inhibitor of IMP dehydrogenase (IMPDH). The inhibition of IMPDH is a new target therapy for glioblastoma. MPA was found to lower TERT levels in both U87MG and U251MG cell lines while also reducing O6-methylguanine-DNA methyltransferase (MGMT) expression in U251 cells [123]. Furthermore, MPA treatment resulted in a decline in telomere repeat numbers in both cell lines. Additionally, MPA induced a dose-dependent increase in p53 and CCCTC-binding factor (CTCF), both known to repress *TERT*, in U87MG and U251 cell lines [123].

## 9. *KMT2C* Gene Mutation and Glioma

The *Lysine Methyltransferase 2* (*KMT2C*) gene, which is known for mixed-lineage leukemia protein 3 (MLL3), mediates the histone methyltransferase activity of Histone H3 on lysine 4 (H3K4) [124]. It controls the regulation of gene expression and DNA accessibility with a strong association with epigenetic modifications [125]. Zhang et al. explored the role of *KMT2C* gene mutations at both the protein level and genetic alterations. They examined 171 genes encoding epigenetic modifier proteins from 283 lower-grade gliomas (LGG). Their findings showed that the *KMT2C* gene was one of the epigenetic modifier genes with mutations and frequent copy number alterations (11%) in lower-grade gliomas and a rare mutation frequency in primary glioblastoma (5%) [126]. Notably, *KMT2C* mutations have been identified in several aggressive cancer types with mutation frequencies, including prostate (6–9%), breast (12%), melanoma (45%), and colon cancer (14%) [124,127,128]. According to the Catalogue Of Somatic Mutations In Cancer (COSMIC), the *KMT2C* gene is mutated in 4% of glioblastomas [95]. In a recent study, the *KMT2C* gene was identified as mutated and was associated with cancer stem cells in four glioblastoma patients using comparative analysis and the exome sequencing [129]. The study was based on the genetic variants of the tumor core and peritumor tissue, and it indicates that the heterozygous indel was recognized at the chr7:151945071 position and was associated with p.Tyr816Ter [129].

Sarker et al. found that glioblastoma-mutated oncodriver receptor genes (derived from astrocyte-like neural stem cells) could be promising drug target candidates [130]. They examined several receptor genes across NCBI whole-exome sequencing (WES) profiles and CGGA databases. They found that the *KMT2C* gene was among the top 20 glioblastoma-mutated genes among other missense-mutated genes. Accordingly, it was defined as one of the 16 glioblastoma top-ranked oncodriver genes derived from the somatically mutated gene set [130]. The *KMT2C* mutations in glioblastoma disrupt DNA damage repair mechanisms, dysregulating genomic stability [131]. Wang et al. demonstrated that the knockout of the *KMT2* gene in urothelial carcinoma resulted in disrupted differentiation and genetic instability, which led to increased inflammatory signals and heightened oncogenic transformation [132].

## 10. *H3F3A* Is a Biomarker for Pediatric Glioma

The *H3F3A* is commonly mutated at two positions (K27M, G34R/G34V) in 31% of pediatric glioblastoma [133]. The H3F3A K27M-mutant pediatric glioblastoma is a distinct entity that transcends the established proneural/mesenchymal/classical adult glioma class. While adult gliomas are subtyped on these lines, with the proneural type having better prognosis and the mesenchymal type being therapy-resistant, pediatric H3K27M-mutant tumors are developing an emerging ‘diffuse midline glioma’ profile with admixed features from these classes [133].

*H3K27M*-mutant tumors have significantly poorer outcomes (12–15-month survival) compared to wildtype pediatric gliomas due to their unfavorable biology and drug resistance [134]. *H3K27M*-mutants are associated with altered H3K27me3 marks and PRC2 complex dysregulation [135]. *H3K27M*-mutant tumors are also resistant to conventional therapies due to their destabilized chromatin structure and stem-cell-like phenotype. Surprisingly, this subtype is differentially sensitive to targeted agents relative to adult gliomas or pediatric H3 wildtype tumors, and thus subtype-specific treatment is needed [136]. Due to the mechanistic complexity of H3F3A, a single treatment option is not recommended; instead, a combination of HDAC inhibitors and radiotherapy has shown promising results in clinical trials. Other clinical trials focus on other pathways like the dopamine receptor DRD2, which promotes tumor growth and stem cell phenotypes in *H3K27M* tumors.

*H3F3A*-mutated pediatric glioblastoma is mechanistically distinct from adult glioblastoma [137]. While adult glioblastoma resistance typically develops from *EGFR* amplification or *MGMT* promoter methylation, pediatric *H3K27M*-mutant tumors evade treatment by employing chromatin remodeling and disrupted developmental processes (e.g., PRC2 dysfunction) [138]. Individually, the *H3F3A* K27M mutation is a vital molecular biomarker of triple clinical significance: (1) it is a diagnostic subtype (diffuse midline glioma), (2) it predicts aggressive clinical behavior and dismal prognosis independent of histologic grade, and (3) it labels tumors with distinct therapeutic vulnerabilities through its epigenetic reprogramming effects [133,134,139,140].

## 11. Wnt Pathway as Glioma Biomarkers

### 11.1. The Wnt Pathway in Glioblastoma

The Wnt/β-catenin signaling pathway is integral to embryogenesis and tissue homeostasis [141]. Signaling permits the regulation of cell proliferation, migration, fate determination, and differentiation [142,143]. Evidence suggests that aberrant activation of this pathway encourages tumorigenesis [143,144].

β-catenin, encoded by *CTNNB1,* plays a dual role in cell maintenance by behaving as a key mediator in intracellular adhesion and an integral downstream regulator of the Wnt/β-catenin pathway [145]. In the presence of Wnt signaling, a Wnt ligand binds to a complex consisting of Frizzled and LRP5/6. Phosphorylation of LRP6 then leads to Disheveled protein recruitment [145]. This protein prevents the destruction complex, consisting of APC, AXIN, CKIa, and GSK-3β, from phosphorylating and ubiquitinating β-catenin. β-catenin can then propagate the cytoplasmic signal into the nucleus [145]. Within the nucleus, β-catenin works directly with transcription factors, such as TCF and LEF [145]. Without the Wnt ligand, the destruction complex phosphorylates β-catenin, which becomes ubiquitinated by SCF, leads to proteasomal degradation [145].

### 11.2. CTNNB1 Mutations Are Rare in Glioblastoma

*CTNNB1* mutations are rare in glioblastoma, unlike other cancers [146]. Mutations in the *CTNNB1* oncogene are present in medulloblastoma, hepatocellular carcinoma, colon and ovarian cancers [147]. In previous studies, fifteen percent of cases analyzed underwent Wnt activation among medulloblastomas, and seventy-three percent of those cases had *CTNNB1* mutations [148]. According to the COSMIC catalog, all medulloblastoma patients displaying a *CTNNB1* mutation within their catalog underwent a missense substitution mutation at the gene [95]. Within the same catalog, out of 5793 hepatocellular carcinoma samples, 20% displayed *CTNNB1* mutations, out of which 83.78% experienced missense substitution mutations, as seen in medulloblastoma patients [95]. The prevalence of *CTNNB1* mutations and their manifestation as missense substitutions is a crucial to factor in when considering how this type of mutation leads to increased proliferation, immunosuppression, and metabolic regulation disruptions, indicating that *CTNNB1* has a role in tumorigenesis [147]. The upregulation of canonical Wnt pathway signaling is associated with mesenchymal glioblastoma, leading to a more motile and invasive behavior compared to proneural or classical subtypes [149,150].

### 11.3. Prognosis

β-catenin mutations promote poor prognosis in neural progenitor cells, based on previous studies on mice conducted by Chenn et al. in 2002 and Zechnar et al. in 2003 [151,152]. Within these studies, a constitutively activated β-catenin causes an increase in proliferative ability in mouse progenitor cells; meanwhile, deletion leads to a decrease in the same cells [153]. Additionally, in biopsies collected from 96 patients, immunohistochemical staining showed several Wnt pathway factors, such as β-catenin, to have protein levels with a positive correlation between the World Health Organization (WHO) glioma grading system and Karnofsky performance scale [154]. Therefore, β-catenin has been considered a potential biomarker in glioblastoma [155,156].

A study conducted by Nager et al. in 2017 found that inhibiting *CTNNB1* in human glioblastoma cell lines, U251MG and U87MG (through shRNA/siRNA), induced autophagy followed by apoptosis [146]. Through this study, it is evident that targeting the CTNNB1/TCF and autophagy pathways behaves as a potential therapeutic option, indicating the importance of β-catenin and, ultimately, *CTNNB1* in glioblastoma cell viability [146].

### 11.4. The Impact of the Wnt Pathway on Drug Efficacy in Glioblastoma

The blockade of the Wnt/β-catenin signal is an active area of research, as it was observed that inhibiting β-catenin in the glioblastoma cell lines, A172, U373, LN18, and LN229 leads to a decrease in the expression of genes associated with proliferation and differentiation. When glioblastoma cell lines were exposed to TMZ and the Wnt/β-catenin inhibitor LGK974, an increase in drug treatment efficacy was observed when compared to TMZ alone [157].

## 12. The NOTCH Pathway Is Commonly Activated in Glioma

The NOTCH signaling pathway is essential for cell differentiation, developing embryos, and tissue homeostasis [158]. The NOTCH pathway uses ligand receptors such as Delta-like-1 and subsequent NOTCH receptor cleavage to send the intracellular domain directly to the nucleus, influencing gene expression [158]. Mechanisms like negative feedback and ubiquitination tightly regulate the NOTCH pathway in a healthy context [158]. This cell-to-cell circuitry allows NOTCH signaling to control key developmental and homeostatic processes like cell maintenance and fate [159]. Notably, NOTCH dysregulation is seen across aggressive cancers such as breast, prostate, and pancreatic cancer, and melanoma [160]. In glioblastoma, NOTCH1 participates in mechanisms such as cell proliferation and increased vascularization [159]. Furthermore, *NOTCH1* mutations are found in approximately 8.5% of gliomas, with higher mutation frequencies in lower-grade gliomas [161,162]. NOTCH1 activation contributes to therapeutic resistance, at least in part, by activating the NFκβ pathway [163]. Moreover, upregulation of the NOTCH1 protein is seen in the clonal cells of glioblastoma, accentuating aggressive local tumor progression and distant relapse [146]. Mutation of *NOTCH1* was found to be associated with a shorter progression-free survival and local relapse in glioblastoma [164]. Current clinical trials include inhibitors of γ-secretase blocking NOTCH signaling [44,45]. Yet, mechanistic value is essential as signal blocking can lead to chemoresistance in some instances.

## 13. Future Directions

At the leading edge of glioma molecular subtyping are evolving omics approaches such as single-cell and spatial transcriptomics, moving quickly into spatial proteomics [15,164,165,166,167]. Transcriptional profiling of single cells and small groups of cells (approximately 50 μm in diameter) has allowed for the sophisticated visualization of glioblastoma tumor heterogeneity. Single-cell sequencing revealed four basic cell states in glioblastoma—neural progenitor-like (NPC), oligodendrocyte progenitor-like (OPC), astrocyte-like (AC), and mesenchymal-like (MES)—and three basic cell states for *IDH*-mutant gliomas: including astrocytic-like, oligodendrocyte-like and undifferentiated, stem cell-like [15,166]. The resolution of spatial transcriptomics is not yet at the single-cell stage but at the level of a small cluster of cells [165,167]. This allows for the investigation of cell–cell contacts within the tumor and with the tumor microenvironment. Proteomics are now also employed in concert with spatial transcriptomics to develop integrative, comprehensive models for how glioma tumors are organized (or not organized in some instances) and interact with the tumor microenvironment [165,167]. Hypoxia drove the organization of cellular states within a tumor in a gradient-like manner in glioblastoma tumors [165]. Hypoxia in glioblastoma is associated with therapeutic resistance and poor immune cell infiltration [168]. Understanding tumor organization, cellular plasticity, and heterogeneity will direct future therapeutic interventions.

## 14. Conclusions

Glioma accounts for 80% of malignant brain tumors and includes glioblastoma, *IDH*-mutant astrocytoma, oligodendroglioma, and ependymoma [1,2,3]. *IDH*-wildtype glioblastoma is associated with poor prognosis (median survival of 14 months) and *PTEN*, *TP53*, *TERT*, *EGFR*, and *PIK3R1* mutations, as shown in Figure 2 [11,31,39,49,50,60,61,75,83,85,99,100,104,117,118,119,121]. Glioblastoma commonly harbors activated PI3K, RAS, Wnt, and NOTCH pathways that drive the highly aggressive phenotype [37,71,147,150,153,163,169,170]. *IDH*-mutant astrocytoma and oligodendroglioma are associated with a better prognosis than glioblastoma, with a median survival of 5–10 years [5,14,18]. *IDH*-mutant gliomas commonly harbor mutations in *PIK3CA*, *TP53*, *CIC*, and 1p/19q co-deletion, Figure 2 [32,56,92]. *IDH*-mutant gliomas have seen promising developments in therapeutics, including the NOA16 phase I clinical trial that employed an IDH1-R132H peptide vaccine that showed strong immunogenicity [25,27]. Although all gliomas are derived from glial cells, these cancers have distinct biomarkers that inform prognosis and therapeutic strategies. Molecular biomarkers for glioma continue to be identified and will be critical in tumor stratification to devise precision therapeutics.

## Figures and Tables

**Figure 1 biomedicines-13-01298-f001:**
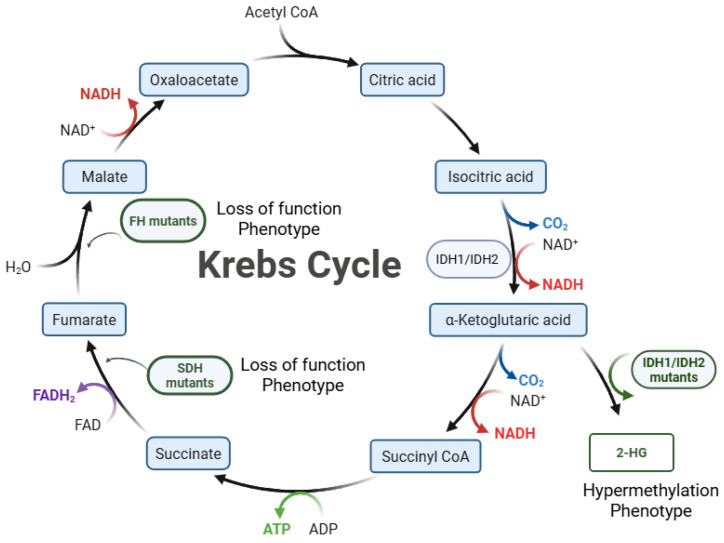
Krebs cycle mutations in glioma. *IDH1/IDH2* mutants convert α-Ketoglutarate to 2-Hydroxyglutarate (2-HG). Loss-of-function mutations in *Succinate Dehydrogenase* (*SDH*) and *Fumarate Hydratase* (*FH*) lead to the accumulation of succinate and fumarate, which inhibit JMJD demethylases to contribute to glioma oncogenesis [8,9].

**Figure 2 biomedicines-13-01298-f002:**
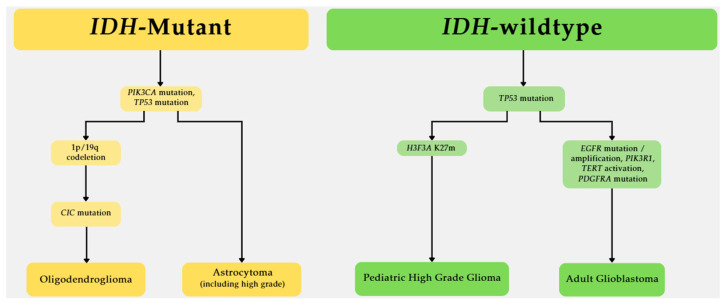
Glioma characteristics associated with *IDH*-mutation status. A flow chart highlights mutations observed in *IDH*-mutant and wildtype gliomas.

**Table 1 biomedicines-13-01298-t001:** Biomarkers with associated drug treatments.

Biomarker	Drug Treatment
**IDH**	IDH1 peptide vaccine [27]
	IDH1 inhibitor ivosidenib [24]
	IDH1/IDH2 inhibitor vorasidenib [24]
**PTEN**	TMZ and poly (ADP-ribose) polymerase (PARP) inhibitors [28]
	IFN-β/TMZ combination treatment [29]
	Bcl-2 inhibitors (e.g., venetoclax, navitoclax, and sabutoclax) [30]
	TNF-α treatment [31]
	p110α inhibitors (e.g., Alpelisib; BYL719) [32]
**PIK3CA**	EGFR inhibitor (AZD-9291), dual PI3K/mTOR inhibitor (GDC-0084) [33]
	Pan-PI3K inhibitor (e.g., Buparlisib; BKM120 and Pilaralisib; XL147), p110α inhibitor (Alpelisib; BYL719), and dual PI3K/mTOR inhibitors (e.g., Dactolisib; NVP-BEZ235 and GDC-0084) [34]
	PI3K inhibitor (GDC-0941) and MEK inhibitor (GDC-0973) [35]
	Dual PI3K/mTOR inhibitors (BEZ235 and BKM120), MEK inhibitors (GSK1120212 and PD0325901) [36]
	mTOR inhibitor (Everolimus), pan-PI3K inhibitor (e.g., BKM120, XL147, PX866, and GDC-0941), p110α inhibitor (such as BYL719), dual PI3K/mTOR inhibitor (such as BEZ235, XL765) [37]
**PIK3R1**	AKT inhibitor (MK2206) [37,38]
	PI3K inhibitors and a combination of PI3K and MAPK signaling inhibition [38]
	[39] MEK inhibitors (PD0325901, AZD6244, GSK1120212B) and JNK inhibitors (SP600125 and BI78D3) [40]
**EGFR**	Small molecule EGFR inhibitors: Afatinib, Dacomitinib, Osimertinib, Rociletinib, Gefitinib, Erlotinib, Lapatinib [41,42,43]
	Monoclonal antibody to EGFR: Cetuximab [41,42,43]
**NOTCH1**	Gamma secretase inhibitors: DAPT and MRK003 [44,45]

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
