# Peer review of "Molecular Biomarkers of Glioma"

_biomedicines, 2025, doi:10.3390/biomedicines13061298_

Round 1

Reviewer 1 Report

Comments and Suggestions for Authors

In third line of introduction, ‘Approximately 51% of primary malignant brain tumors are classified as glioblastoma’, please explain that is it 51% of all kind of tumors or 51% of brain tumors? The reference 1 from where this sentence is derived, seems to mention that it is 51% of all malignant tumors.

‘In glioblastoma, PTEN mutations are frequently found in primary tumors, approximately 30%, compared to [23], secondary tu-mors, which were found to be only 4% [24,25]..’

Comment: in this sentence, it is 30% mentioned but in reference 23, 8% is mentioned for the contribution of PTEN mutations . In reference 25 (1997 study), the percentage of PTEN mutations in secondary tumors is not 4%, its approx. 60%. Please revisit

A recent study published in 2025 has reported ATRX mutation along Ras pathway (reference below)

Yu R, Huang K, He X, Zhang J, Ma Y, Liu H. ATRX mutation modifies the DNA damage response in glioblastoma multiforme tumor cells and enhances patient prognosis. Medicine (Baltimore). 2025 Jan 10;104(2):e41180. doi: 10.1097/MD.0000000000041180. 

Classification of biomarkers should also be revisited. At some places they are mentioned according to pathway but at other places it is mentioned according to mutation. E.g. Ras pathway and Capicua Transcriptional Repressor (its MAP kinase pathway).

A summarized table should be incorporated with all these biomarkers for an easy understanding

A flow chart of classification of biomarkers can be added

Author Response

In third line of introduction, ‘Approximately 51% of primary malignant brain tumors are classified as glioblastoma’, please explain that is it 51% of all kind of tumors or 51% of brain tumors? The reference 1 from where this sentence is derived, seems to mention that it is 51% of all malignant tumors.

We reworded this sentence to be clearer: Glioblastoma accounts for 50% of primary, malignant brain tumors and has a median survival of 12-14 months. (page 1)

‘In glioblastoma, PTEN mutations are frequently found in primary tumors, approximately 30%, compared to [23], secondary tu-mors, which were found to be only 4% [24,25]..’

We corrected typos (page 4).

Comment: in this sentence, it is 30% mentioned but in reference 23, 8% is mentioned for the contribution of PTEN mutations . In reference 25 (1997 study), the percentage of PTEN mutations in secondary tumors is not 4%, its approx. 60%. Please revisit

We revisited and revised this section.

In glioblastoma, PTEN mutations are frequently found in primary tumors, approximately 32%, compared secondary tumors, which were found to be only 4% [24,25]. Fixed on page 4.

A recent study published in 2025 has reported ATRX mutation along Ras pathway (reference below)

Yu R, Huang K, He X, Zhang J, Ma Y, Liu H. ATRX mutation modifies the DNA damage response in glioblastoma multiforme tumor cells and enhances patient prognosis. Medicine (Baltimore). 2025 Jan 10;104(2):e41180. doi: 10.1097/MD.0000000000041180. 

We added this helpful information to the Ras section of our manuscript (page 7).

Add CIC to Ras Classification of biomarkers should also be revisited. At some places they are mentioned according to pathway but at other places it is mentioned according to mutation. E.g. Ras pathway and Capicua Transcriptional Repressor (its MAP kinase pathway).

 We made these changes (pages 8-9) .

A summarized table should be incorporated with all these biomarkers for an easy understanding

We include a table with biomarkers and associated therapies (pages 14-15, requested by another reviewer).

A flow chart of classification of biomarkers can be added

We include a flow chart (page 12).

Reviewer 2 Report

Comments and Suggestions for Authors

This review discusses a wide range of molecular biomarkers involved in glioma development, classification, prognosis, and treatment. The manuscript thoroughly integrates recent literature and emphasizes both the biological mechanisms and clinical implications of key mutations such as IDH1/2, PTEN, TP53, TERT, CIC, and components of the PI3K, RAS, Wnt, and NOTCH pathways. The inclusion of clinical trial data and therapeutic prospects significantly adds to the translational value of this work. However, several areas require improvement. Below are detailed comments and suggestions to improve the manuscript.

1. The second section provides a discussion on the molecular characteristics of IDH-mutant lower-grade gliomas. recommend to cite PMID: 38971948 to highlight the role of miRNA-mediated regulatory mechanisms in lower-grade glioma.

2. recommend to add "Future Research Directions" section, briefly outlining how high-throughput single-cell sequencing and spatial transcriptomics can aid in deciphering the heterogeneity of gliomas, and how integrative multi-omics analyses may enhance molecular subtyping and precision therapy.

3. Section 13 could be further strengthened by integrating recent evidence on its involvement in clonal evolution and distant relapse, such as PMID: 37649695, which highlight NOTCH signaling as a key driver of local tumor progression and recurrence. 

4. There are some grammatical errors throughout the manuscript.  Page 1, Abstract:"astroytoma" →  "astrocytoma"

5. The discussion of clinical significance is not sufficiently in-depth. In most sections, the manuscript merely reports the mutation frequency of specific genes and their association with prognosis, but lacks further exploration of their clinical translational value. It is recommended to expand the discussion to address questions such as: Can certain biomarkers be detected preoperatively? Are they applicable for liquid biopsy? What is the current level of recommendation for these biomarkers in major clinical guidelines (e.g., NCCN, WHO CNS 5th edition)?

Author Response

This review discusses a wide range of molecular biomarkers involved in glioma development, classification, prognosis, and treatment. The manuscript thoroughly integrates recent literature and emphasizes both the biological mechanisms and clinical implications of key mutations such as IDH1/2, PTEN, TP53, TERT, CIC, and components of the PI3K, RAS, Wnt, and NOTCH pathways. The inclusion of clinical trial data and therapeutic prospects significantly adds to the translational value of this work. However, several areas require improvement. Below are detailed comments and suggestions to improve the manuscript.

  1. The second section provides a discussion on the molecular characteristics of IDH-mutant lower-grade gliomas. recommend to cite PMID: 38971948 to highlight the role of miRNA-mediated regulatory mechanisms in lower-grade glioma.

We added this helpful citation (page 3).

  1. recommend to add "Future Research Directions" section, briefly outlining how high-throughput single-cell sequencing and spatial transcriptomics can aid in deciphering the heterogeneity of gliomas, and how integrative multi-omics analyses may enhance molecular subtyping and precision therapy.

We added this section (page 15).

  1. Section 13 could be further strengthened by integrating recent evidence on its involvement in clonal evolution and distant relapse, such as PMID: 37649695, which highlight NOTCH signaling as a key driver of local tumor progression and recurrence. 

We added this helpful citation (page 14).

  1. There are some grammatical errors throughout the manuscript.  Page 1, Abstract:"astroytoma" →  "astrocytoma"

Thank you. We corrected typos.

  1. The discussion of clinical significance is not sufficiently in-depth. In most sections, the manuscript merely reports the mutation frequency of specific genes and their association with prognosis, but lacks further exploration of their clinical translational value. It is recommended to expand the discussion to address questions such as: Can certain biomarkers be detected preoperatively? Are they applicable for liquid biopsy? What is the current level of recommendation for these biomarkers in major clinical guidelines (e.g., NCCN, WHO CNS 5th edition)?

We added this revision to most of the sections (pages 2-3, page 4, page 5, page 6, page 7, page 8, page 9, and page 11).

Reviewer 3 Report

Comments and Suggestions for Authors The manuscript by Megan Keniry et al., entitled "Molecular biomarkers of Glioma" provides a comprehensive review of various biomarkers associated with glioma. Addressing some minor comments will further improve the manuscript.    Kindly include a figure of the Krebs cycle highlighting the various proteins associated with glioma   Kindly include a table summarizing the various glioma biomarkers, their drug treatments mentioned in the manuscript, along with the corresponding references cited.

Author Response

Comments and Suggestions for Authors

The manuscript by Megan Keniry et al., entitled "Molecular biomarkers of Glioma" provides a comprehensive review of various biomarkers associated with glioma. Addressing some minor comments will further improve the manuscript.   

Kindly include a figure of the Krebs cycle highlighting the various proteins associated with glioma  

We include this as Figure 1 (page 2).

Kindly include a table summarizing the various glioma biomarkers, their drug treatments mentioned in the manuscript, along with the corresponding references cited.

We include this as Table 1 (page 14).

Round 2

Reviewer 1 Report

Comments and Suggestions for Authors

manuscript is significantly improved and revised

Author Response

R1: manuscript is significantly improved and revised

Thank you for reviewing our manuscript.

Reviewer 2 Report

Comments and Suggestions for Authors

Authors have made substantial improvements to the manuscript. The revised version is more robust in methodological clarity, and interpretation of findings.

Suggestions for final improvement:

The expression "IDH1 (Isocitrate dehydrogenase 1)" is not standard and should be corrected to "IDH1 (isocitrate dehydrogenase 1)".

Authors may consider incorporating recent insights into IDH1 and IDH2 mutations in glioma, such as PMID: 38321118. Integrating this perspective in the "IDH1 Mutations" sections—would enhance the mechanistic depth and align with the emphasis on molecular biomarkers.

"fumerate" → "fumarate" (legend of Figure 1)

Author Response

R2: Authors have made substantial improvements to the manuscript. The revised version is more robust in methodological clarity, and interpretation of findings.

Suggestions for final improvement:

The expression "IDH1 (Isocitrate dehydrogenase 1)" is not standard and should be corrected to "IDH1 (isocitrate dehydrogenase 1)".

We made this change.

Authors may consider incorporating recent insights into IDH1 and IDH2 mutations in glioma, such as PMID: 38321118. Integrating this perspective in the "IDH1 Mutations" sections—would enhance the mechanistic depth and align with the emphasis on molecular biomarkers.

Thank you for this insightful suggestion. We include discussion from PMID: 38321118.

"fumerate" → "fumarate" (legend of Figure 1)

We made this change. Thank you for reviewing our manuscript.